# Progressive Increase in Small Intestinal Bacterial Overgrowth Risk Following COVID-19 Infection: A Global Population-Based Study

**DOI:** 10.3390/diseases13090275

**Published:** 2025-08-22

**Authors:** Yilin Song, Thai Hau Koo, Benjamin D. Liu, Linda L. D. Zhong, Tao Bai, Xiaohua Hou, Lei Tu, Gengqing Song

**Affiliations:** 1Department of Internal Medicine, University of Maryland Medical Center Midtown Campus, Baltimore, MD 21201, USA; yilin.song@umm.edu; 2Department of Endocrinology Research Unit, Mayo Clinic, Rochester, MN 55902, USA; 3Gastrointestinal Function and Motility Unit, Division of Gastroenterology and Hepatology, Department of Internal Medicine, University Sains Malaysia Specialist Hospital, Kelantan 16150, Malaysia; waynehau@usm.my; 4Department of Internal Medicine, MetroHealth Medical Center, Case Western Reserve University, Cleveland, OH 44139, USA; benjamin.liu1@nm.org; 5School of Biological Sciences, Nanyang Technological University, Singapore 637551, Singapore; linda.zhong@ntu.edu.sg; 6Division of Gastroenterology, Union Hospital, Tongji Medical College, Huazhong University of Science and Technology, Wuhan 430022, China; baitao@hust.edu.cn (T.B.); houxh@hust.edu.cn (X.H.); 7Department of Gastroenterology and Hepatology, MetroHealth Medical Center, Case Western Reserve University, Cleveland, OH 44139, USA

**Keywords:** small intestinal bacterial overgrowth (SIBO), coronavirus disease 2019 (COVID-19), gut microbiota, renin–angiotensin system, TriNetX database

## Abstract

Background/Objectives: Coronavirus disease 2019 (COVID-19) is associated with gastrointestinal (GI) symptoms. Small intestinal bacterial overgrowth (SIBO) is emerging as a significant GI sequela post-COVID-19 infection. This study aimed to evaluate the prevalence and incidence of SIBO post-COVID-19 infection across different age groups and to identify associated risk factors in a global cohort. Methods: A retrospective study utilized the TriNetX database and included adult patients (≥18 years) diagnosed with SIBO following COVID-19 infection (1 January 2022–30 May 2024). A propensity score matching (1:1) was used to adjust for demographics and SIBO risk factors. Kaplan–Meier survival analysis assessed the SIBO incidence within 12 months. Results: Among 1,660,092 COVID-19 patients and 42,322,017 controls, SIBO was diagnosed in 353 COVID-19 patients without hydrogen breath tests (BT) and 78 with BT, compared to 3368 controls without BT and 871 with BT. Age-specific analysis demonstrated a clear, progressive increase in the SIBO incidence, becoming distinctly significant by 6 months and highest at 12 months post-infection. The highest risks were noted in ages 60–69 (0.011% vs. 0.004%, OR 2.6, *p* = 0.0003) and 70–79 (0.011% vs. 0.005%, OR 2.0, *p* = 0.0004), with younger age groups (30–49 years) also showing significantly increased risks. The medication analysis revealed strong associations with chronic opioid, proton pump inhibitor, and antidiarrheal medication. Conclusions: COVID-19 significantly increased the risk of SIBO, particularly within the first 12 months post-infection, across various age groups and, notably, in association with certain chronic medications. Clinical vigilance and targeted management strategies are recommended to mitigate long-term GI consequences.

## 1. Introduction

Coronavirus disease 2019 (COVID-19) is associated with a group of gastrointestinal (GI) symptoms, including abdominal pain, vomiting, diarrhea, a loss of appetite, acid reflux, and esophageal disorders during and after the acute phase of COVID-19 infection [1,2,3]. Small intestinal bacterial overgrowth (SIBO) is not a common GI symptom caused by COVID-19 infection and there are only a handful of reports on new-onset SIBO cases post-infection [4,5,6]. Possible mechanisms include gut microbiota disturbance and renin–angiotensin system dysfunction. Previous studies have shown fecal microbiome changes up to 30 days post-COVID-19 infection [7,8,9].

The COVID-19 virus is believed to bind to the angiotensin-converting enzyme 2 (ACE-2) receptor on the alveolar epithelial cells, leading to lung infection. It is hypothesized that the virus also binds to the ACE-2 receptor in the small intestine epithelia cells, causing an imbalance in the gut microbiota [10,11,12]. Furthermore, a population-based study showed gastrointestinal motility disorders lasting up to one year post-COVID-19 infection, which may contribute to the development of SIBO [13].

SIBO is caused by an excessive bacteria presence in the small intestine and is associated with a range of symptoms, including bloating, abdominal distension, abdominal pain/discomfort, diarrhea, fatigue, and weakness [14]. Patients with SIBO can develop severe malnutrition, vitamin B12 deficiency, anemia, osteoporosis, and steatorrhea, which can lead to more severe adverse outcomes or the worsening of comorbidities without timely treatment [14]. Breath tests (BT) of lactulose and glucose are recommended for SIBO diagnosis in symptomatic patients [15]. Antibiotic treatment is warranted upon diagnosis and can reduce the recurrence of SIBO [16].

Given the lack of data on the prevalence and incidence of SIBO following COVID-19 infection in large population-based cohorts, the aim of this study was to determine whether the risk of SIBO development increased after COVID-19 infection. The correlations between clinical characteristics, comorbidities, medications, and SIBO prevalence, as well as the percentage of SIBO cases diagnosed with BT, were assessed as secondary outcomes.

## 2. Materials and Methods

A retrospective study was conducted with the TriNetX database (Cambridge, MA, USA), a de-identified, multicenter global database that provides anonymized real-world data. It encompasses over 275 million patient records from 152 healthcare organizations across 21 countries [17]. The platform enables patient-level analysis while preserving anonymity—any data points involving 10 or fewer individuals are not displayed, ensuring compliance with the strict privacy standards. The Institutional Review Board of the University of Maryland and MetroHealth Medical Center classified studies using TriNetX as exempt from the IRB review, as the data meet the de-identification criteria outlined in Section §164.514 (a) of the HIPAA Privacy Rule. This study was reported in accordance with the STROBE (Strengthening the Reporting of Observational Studies in Epidemiology) guidelines for cohort studies [18].

### 2.1. Patient Selection

Adult patients aged 18 years or older from 1 January 2022 to 30 May 2024 were identified using the TriNetX database. The time frame was selected to align with the global adoption of SIBO diagnostic coding and a 12-month follow-up period. The COVID-19 cohort included patients diagnosed with COVID-19 or those with a recorded positive COVID-19 test result. The control cohort excluded patients with any documented COVID-19 diagnosis or positive test result (Figure 1). Both cohorts were subject to the same exclusion criteria, including a history of major GI or abdominal surgeries (e.g., bariatric surgery and cesarean sections), connective tissue disorders (e.g., amyloidosis and Ehlers–Danlos syndrome), Sjögren’s syndrome, Parkinson’s disease, inflammatory bowel diseases (e.g., ulcerative colitis and Crohn’s disease), and other GI conditions such as cystic fibrosis and esophageal dyskinesia (Appendix A Table A1).

### 2.2. Query Validation

Queries were validated by performing an unmatched analysis on patients diagnosed with SIBO, with or without COVID-19 infection from 1 January 2022 to 30 May 2024, to align with the global adoption of the SIBO diagnostic coding time and 12-month follow-up period. From the cohorts of patients with SIBO, we further identified the number of patients who underwent hydrogen BT.

### 2.3. Covariates and Matching

Confounding factors that could influence the risk of SIBO were identified. One-to-one propensity score matching (PSM) was performed using these factors, which included demographic variables such as the age at index event, sex, ethnicity, race, body mass index (BMI), end-stage renal disease, and medication including opioids, diphenoxylate, and proton pump inhibitors (PPIs).

### 2.4. Propensity-Matched Study Outcomes

The outcome of this study was the development of SIBO. This outcome was analyzed at 1, 3, 6, and 12 months post-COVID-19 infection. The two cohorts were further stratified with the age groups: 18–29, 30–39, 40–49, 50–59, 60–69, 70–79, and ≥80 years. The risk of SIBO development was compared among all age groups at each of the four follow-up time points.

### 2.5. Statistical Analysis

All statistical analyses were conducted using the TriNetX platform [17]. Outcomes were summarized using means, standard deviations (SD), and percentages, as appropriate. PSM was performed in a 1:1 fashion using relevant covariates via a greedy nearest neighbor algorithm with a caliper width of 0.1. Matched cohorts were considered well-balanced based on baseline characteristics. Kaplan–Meier survival analysis was conducted within the TriNetX platform. Patients were followed until the occurrence of a SIBO outcome, their last recorded clinical encounter, or death—whichever occurred first. Odds ratios (ORs) with 95% confidence intervals (CIs) were calculated using the TriNetX analytics platform to compare SIBO outcomes between cohorts. Statistical significance was defined as a *p*-value less than 0.05. The figure was generated in Microsoft PowerPoint (Redmond, WA, USA) version 2505.

## 3. Results

### 3.1. Overall Prevalence and Incidence of SIBO

A total of 1,660,092 COVID-19 patients and 42,322,017 controls were analyzed. SIBO was diagnosed in 353 (0.021%) COVID-19 patients without hydrogen BT and 78 (0.005%) patients confirmed by BT, compared to 3368 (0.008%) controls without BT and 871 (0.002%) with BT (Table 1).

### 3.2. Demographic Characteristics

COVID-19 patients diagnosed with SIBO were significantly older (51.4 ± 17.3 years vs. 47.3 ± 17.5 years, *p* < 0.001) and had a higher BMI (26.5 ± 6.0 kg/m^2^ vs. 25.6 ± 5.8 kg/m^2^, *p* = 0.008) compared to controls. Female patients constituted approximately 75% in both groups, without a significant difference (*p* = 0.855). The ethnic composition showed that COVID-19 patients with SIBO had slightly fewer Hispanic or Latino patients (4.8% vs. 6.3%, *p* = 0.071) and more White individuals (82.4% vs. 75.8%, *p* = 0.174). Statistically significant differences were observed among Black or African American (6.5% vs. 5.9%, *p* = 0.030) and Asian patients (2.8% vs. 3.6%, *p* = 0.048, Table 1).

### 3.3. Age-Specific Incidence of SIBO

The incidence of SIBO showed a progressive and time-dependent increase following COVID-19 infection. While no significant differences between the COVID-19 and control cohorts were observed at 1-month post-infection, a trend toward higher risk began to emerge at 3 months, which was particularly noticeable in the 70–79 years age group (*p* = 0.012). By 6 months, this elevated risk became more apparent and statistically significant across multiple age groups (ages 30–69; all *p* < 0.05, Table 2).

At the 12-month follow-up, the increased risk was most pronounced and statistically significant across all age groups studied. Specifically, older adults (ages 60–69 and 70–79) exhibited the highest incidences of SIBO (0.011% in COVID-19 patients compared to 0.004–0.005% in controls; OR 2.6, 95% CI 1.3–5.4, *p* = 0.0003, and OR 2.0, 95% CI 0.94–4.27, *p* = 0.0004, respectively; Table 3).

Significant elevations were also evident in younger cohorts: ages 30–39 years (0.009% vs. 0.003%; OR 2.7, 95% CI 1.3–5.8, *p* = 0.002) and 40–49 years (0.009% vs. 0.004%; OR 2.4, 95% CI 1.1–5.0, *p* < 0.0001; Table 3). Additionally, in patients aged ≥ 80 years, a noteworthy incidence of 0.007% was observed, whereas no cases were reported in the matched control group (*p* = 0.021).

### 3.4. Medication Use and Associated Risks

Significant medication use associated with an increased risk of SIBO among COVID-19 patients included chronic opioid use (3% vs. 0.4%, *p* < 0.001) and proton pump inhibitors (PPIs) such as pantoprazole (18% vs. 12%, *p* = 0.002) and omeprazole (32% vs. 18%, *p* < 0.001). Antidiarrheal medications were also associated with an increased risk, including loperamide (3% vs. 1%, *p* < 0.001) and diphenoxylate (3% vs. 0%, *p* < 0.001; Table 4).

## 4. Discussion

COVID-19 has been widely associated with GI symptoms, including bloating, gas, abdominal pain, and discomfort, as well as significant long-term disruptions to the gut microbiome [1,11,13]. Although isolated case reports have previously identified instances of SIBO following COVID-19 [4,5,6,19], our study is the first to quantitatively analyze the incidence of SIBO using extensive global real-world data. Our findings indicate a notable, time-dependent increase in the risk of SIBO, becoming particularly evident by 6 months and peaking at 12 months post-infection. This comprehensive analysis substantially advances our understanding of post-COVID-19 gastrointestinal outcomes, highlighting both age-specific vulnerabilities and medication-related risk factors.

The development of SIBO typically occurs due to disruptions in the normal gut bacterial equilibrium, often resulting from diminished gastric acid secretion or impaired intestinal motility [14]. COVID-19 patients exhibit sustained alterations in their fecal microbiome for up to 30 days post-infection [7,8,9]. Large-scale analyses, such as those based on the U.S. Department of Veterans Affairs database, have similarly highlighted persistent GI complications, including motility disorders, lasting up to one year post-infection [13]. Supporting studies have further documented similar gastrointestinal disruptions at shorter intervals, including 90 days and 6 months post-infection [3,20]. These previous observations closely align with our findings, which demonstrate a clear progressive increase in SIBO incidence, likely attributable to microbiome imbalances and motility dysfunctions. A previous study involving 495 adult hospitalized patients with COVID-19 in Italy indicated differences in the age, comorbidities, and hospital stay length between sexes [21]. Our data did not demonstrate gender differences in these aspects, potentially due to the difference in size, the geography of the study population, and the fact that we included both the inpatient and outpatient population.

Prior studies have reported a SIBO prevalence ranging broadly from 2.5% to 22% across various populations [22], with significantly higher rates, up to 93.8%, observed among patients with post-COVID-19 irritable bowel syndrome [23]. Recognizing the challenge of precisely attributing the SIBO etiology, our study applied rigorous exclusion criteria and PSM to minimize confounding factors. Notably, the SIBO prevalence is reported to be particularly high, up to 52.5%, among individuals aged over 75, roughly twice the prevalence in younger populations, highlighting potential under-diagnosis in older demographics. Our findings reinforce this observation, demonstrating a significantly increased SIBO risk among individuals aged over 80 within 12 months post-COVID-19, compared to matched controls. Given that untreated SIBO can lead to severe nutritional and metabolic complications, including malnutrition, vitamin B12 deficiency, anemia, osteoporosis, and steatorrhea [14], timely diagnosis and intervention are crucial. Moreover, while older adults (60–79 years) displayed the highest incidence, the significant SIBO risk identified among younger individuals (30–49 years) underscores the necessity of broad clinical vigilance across all age groups.

In our study, the majority of SIBO diagnoses among COVID-19 patients were based on clinical presentations rather than BT confirmations, reflecting limitations in current diagnostic practices. The American College of Gastroenterology (ACG) guidelines recommend BT primarily for symptomatic patients with IBS, suspected motility disorders, or a history of luminal abdominal surgery, and advise against its routine use in asymptomatic patients on PPIs due to limited evidence [15]. BT’s diagnostic utility remains controversial due to relatively low sensitivity (42% for lactulose and 54.5% for glucose) and potential false positives driven by colonic fermentation rather than true SIBO [24,25]. The significant increase in the post-COVID-19 SIBO incidence observed in our study emphasizes the critical need for enhanced clinical vigilance, improved diagnostic methodologies, and proactive therapeutic interventions.

Another significant observation was the increased SIBO risk associated with chronic opioid, diphenoxylate, and PPI usage. Both opioids and diphenoxylate, acting as µ receptor agonists, can significantly slow GI transit and disrupt the gut microbiota composition [26,27]. Similarly, long-term PPI use increases the risk of SIBO by reducing gastric acid secretion, a relationship consistently demonstrated in multiple meta-analyses [28,29]. Our findings, which align closely with these previously reported associations, reinforce the need for the cautious prescribing and optimized clinical management of PPIs and antiperistaltic agents to minimize the potential risk of SIBO.

There are several limitations of this study. As a retrospective cohort study, it carries limitations such as variability in the data quality and the researchers’ lack of control over how the data were originally collected. Additionally, the potential misdiagnosis and incomplete electronic health records (EHR) can interfere with the result accuracy, especially among countries/areas utilizing physical paper format medical records. Another limitation of utilizing this database is that the large study population size may lead to the overestimation of the statistical significance. The under-detection of COVID-19 is another limitation, especially among asymptomatic patients or those with mild upper respiratory symptoms. This concern also applies to the under-detection of SIBO, where patients with a younger age or mild symptoms might not seek medical care. SIBO cases identified in both COVID-19 and non-COVID-19 cohorts were less than 10 at multiple follow-up times, which could be caused by the under-detection of COVID-19 and SIBO cases. Furthermore, we could not collect the information on whether the study population consumed probiotic or other types of supplement that might alter their gut microbiota environment. To further validate and extend our findings, prospective studies are essential. Prospective designs would ensure higher data accuracy and quality control. The further refinement of diagnostic methodologies, such as revising hydrogen BT cutoff thresholds to enhance sensitivity [30], and investigations into antibiotic management strategies for SIBO are also promising avenues for future research. Understanding the mechanisms underlying COVID-19′s impact on SIBO development would further enhance the clinical relevance and applicability of our findings.

## 5. Conclusions

This study provides novel and robust evidence of a significant and progressive increase in the SIBO incidence following COVID-19 infection using a global, multicenter, real-world dataset. The elevation of the SIBO risk was most pronounced by the 12-month follow-up, prominently affecting older adults aged 60–79 years, with notable implications for younger populations as well. These findings highlight the importance of the increased clinical awareness, early diagnosis, and proactive management of SIBO in patients recovering from COVID-19, particularly among elderly individuals and those utilizing chronic medications.

## Figures and Tables

**Figure 1 diseases-13-00275-f001:**
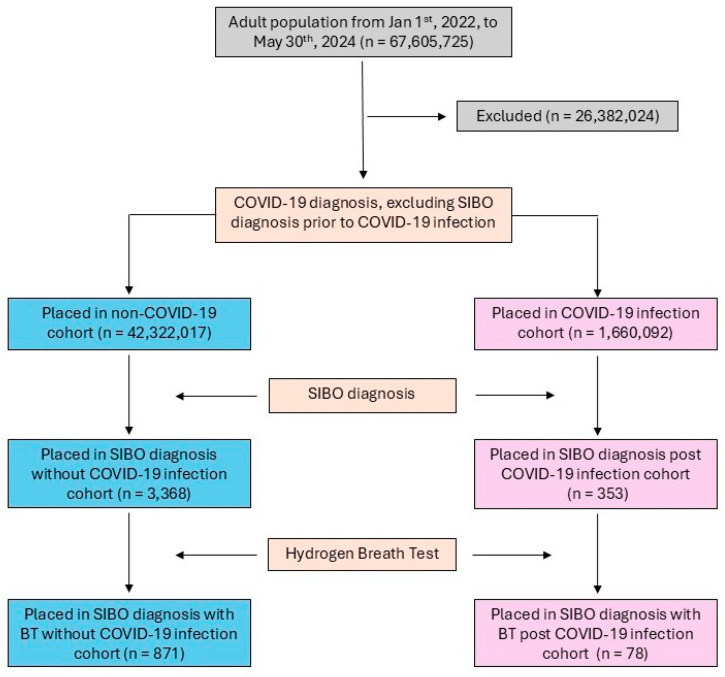
CONSORT diagram of COVID-19/control cohorts’ selection flow.

**Table 1 diseases-13-00275-t001:** Demographic and comorbidities of all study populations and cohorts of SIBO patients with or without COVID-19 infection.

Groups	All Study Population (Exclude SIBO History)		Adults with SIBO Diagnosis AND Hydrogen Breath Test			Adults with SIBO Diagnosis without Hydrogen Breath Test		
	COVID-19 Infection	Non-COVID-19 Infection *	COVID-19 Infection	Non-COVID-19 Infection	*p* Value	COVID-19 Infection	Non-COVID-19 Infection	*p* Value
Total number of cohort population (SIBO prevalence)	1,660,092	42,322,017	78 (0.005%)	871 (0.002%)	N/A	353 (0.021%)	3368 (0.008%)	N/A
Demographics								
Age (years)	50.9 ± 19.7	50.2 ± 19.4	46.6 ± 17.2	44.4 ± 17.3	0.234	51.4 ± 17.3	47.3 ± 17.5	<0.001
BMI (kg/m^2^)	28.5 ± 7.3	27.7 ± 7.1	27.4 ± 5.6	25.7 ± 5.7	0.007	26.5 ± 6.0	25.6 ± 5.8	0.008
% Female	55.4	54.1	66.7	74.2	0.104	75.4	74.9	0.855
% Hispanic or Latino	6.6	6.2	11.2	6.8	0.105	4.8	6.3	0.071
% White	50.8	44.3	76.9	72.7	0.223	82.4	75.8	0.174
% Black or African American	12.8	9.9	10.2	5.4	0.415	6.5	5.9	0.030
% Asian	5.6	4.2	10.4	4.3	0.170	2.8	3.6	0.048
Comorbidities								
% Respiratory ventilation requirement < 24 h	0.060	0.002	0	0	N/A	0	0	N/A
% Respiratory ventilation requirement 24–96 h	0.080	0.002	0	0	N/A	0	0	N/A
% End-stage renal disease	0.039	0.001	0	1.1	0.276	0	0.3	0.305
% Common variable immunodeficiency	0.012	0.0003	0	0	N/A	3	0.3	<0.001
% AIDS	0.009	0.0002	0	0	N/A	0	0	N/A

Abbreviations: small intestinal bacterial overgrowth, SIBO; coronavirus disease 2019 (COVID-19). * Unable to analyze *p* value due to large sample size per TriNetX limitation.

**Table 2 diseases-13-00275-t002:** Risk of SIBO within 1, 3, 6, and 12 months after COVID-19 infection versus without COVID-19 infection in different age groups.

Follow Up Time		Adults with SIBO Diagnosis Post COVID-19 Infection (% Cases in Cohort)				Adults with SIBO Diagnosis Without COVID-19 Infection (% Cases in Cohort)			
		1 Month	3 Months	6 Months	12 Months	1 Month	3 Months	6 Months	12 Months
**Age Group (years)**	18–29	0.004%	0.004%	0.004%	0.005%	0.000% **	0.004%	0.004%	0.003%
	30–39	0.003%	0.003%	0.003%	0.009%	0.000% **	0.003%	0.003%	0.003% **
	40–49	0.003%	0.003%	0.003%	0.009%	0.000% **	0.003%	0.003%	0.004% *
	50–59	0.004%	0.004%	0.005%	0.009%	0.004%	0.004%	0.004%	0.004% *
	60–69	0.000%	0.004%	0.004%	0.011%	0.004% **	0.004%	0.004%	0.004% **
	70–79	0.005%	0.005%	0.005%	0.011%	0.000% **	0.000% **	0.005%	0.005%
	≥80	0.007%	0.007%	0.007%	0.007%	0.000% **	0.000% **	0.000% **	0.000% **

Abbreviations: small intestinal bacterial overgrowth, SIBO; coronavirus disease 2019 (COVID-19). * *p* < 0.05 compared to COVID-19 cohort; ** *p* < 0.01 compared to COVID-19 cohort.

**Table 3 diseases-13-00275-t003:** SIBO cases within 1, 3, 6, and 12 months after COVID-19 infection versus without COVID-19 infection post-PSM in different age groups.

**Age Groups (Years)**	**Number of Pairs**	**1 Month**					**3 Months**				
		**COVID-19**	**Non-COVID-19**	**OR**	**95% CI**	***p* ** **Value**	**COVID-19**	**Non-COVID-19**	**OR**	**95% CI**	***p* ** **Value**
18–29	277,530	≤10	0	NA	NA	0.334	≤10	≤10	1.0	0.416, 2.403	0.733
30–39	291,117	≤10	0	NA	NA	0.175	≤10	≤10	1.0	0.416, 2.403	0.376
40–49	257,178	≤10	0	NA	NA	0.057	≤10	≤10	1.0	0.416, 2.403	0.133
50–59	242,131	≤10	≤10	1.0	0.416, 2.403	0.981	≤10	≤10	1.0	0.416, 2.403	0.048
60–69	245,034	0	≤10	NA	NA	0.297	≤10	≤10	1.0	0.416, 2.403	0.619
70–79	187,844	≤10	0	NA	NA	0.096	≤10	0	NA	NA	0.012
>80	147,915	≤10	0	NA	NA	0.185	≤10	0	NA	NA	0.104
**Age groups (Years)**	**Number of Pairs**	**6 Months**					**12 Months**				
		**COVID-19**	**Non-COVID-19**	**OR**	**95% CI**	***p* ** **Value**	**COVID-19**	**Non-COVID-19**	**OR**	**95% CI**	***p* ** **Value**
18–29	277,530	≤10	≤10	1.0	0.416, 2.403	0.878	13	≤10	1.3	0.570, 2.965	0.645
30–39	291,117	≤10	≤10	1.0	0.416, 2.402	0.014	27	≤10	2.7	1.307, 5.578	0.002
40–49	257,178	≤10	≤10	1.0	0.416, 2.403	0.013	24	≤10	2.4	1.148, 5.019	<0.0001
50–59	242,131	11	≤10	1.1	0.467, 2.590	0.007	22	≤10	2.2	1.042, 4.646	0.0006
60–69	245,034	11	≤10	1.1	0.467, 2.590	0.024	26	≤10	2.6	1.254, 5.392	0.0003
70–79	187,844	≤10	≤10	1.0	0.416, 2.403	0.084	20	≤10	2.0	0.936, 4.273	0.0004
>80	147,915	≤10	0	NA	NA	0.060	≤10	0	NA	NA	0.021

Abbreviations: Propensity Score Matching, PSM; small intestinal bacterial overgrowth, SIBO; coronavirus disease 2019 (COVID-19); odd ratio (OR); confidence intervals (CI). The *p* values were from Kaplan–Meier analysis on SIBO risk of COVID-19 patients compared to non-COVID-19 population. All cases less than 10 were rounded to 10 to protect patient privacy by TriNetX.

**Table 4 diseases-13-00275-t004:** Medication use and associated risks of SIBO among population with or without COVID-19 infection.

Groups	All Study Population (Exclude SIBO History)		Adults with SIBO Diagnosis AND Hydrogen Breath Test			Adults with SIBO Diagnosis WITHOUT Hydrogen Breath Test		
	COVID-19 Infection	Non-COVID-19 Infection *	COVID-19 Infection	Non-COVID-19 Infection	*p* Value	COVID-19 Infection	Non-COVID-19 Infection	*p* Value
Total number of cohort population (SIBO prevalence)	1,660,092	42,322,017	78 (0.005%)	871 (0.002%)	N/A	353 (0.021%)	3368 (0.008%)	N/A
Medications								
% Chronic opioid use	0.094	0.004	10	1	<0.001	3	0.4	<0.001
% Loperamide	0.043	0.001	0	1	0.276	3	1	<0.001
% Dicyclomine	0.029	0.001	10	4	0.021	7	3	0.001
% Diphenoxylate	0.009	0.0001	10	1	<0.001	3	0	<0.001
% Pantoprazole	2.635	0.101	17	12	0.095	18	12	0.002
% Omeprazole	3.010	0.136	40	26	0.003	32	18	<0.001
% Lansoprazole	0.710	0.021	10	2	<0.001	3	2	0.290
% Rabeprazole	0.121	0.008	0	1	0.276	3	1	<0.001
% Esomeprazole	0.674	0.034	10	4	0.004	5	3	0.064

Abbreviations: small intestinal bacterial overgrowth, SIBO; coronavirus disease 2019 (COVID-19). * Unable to analyze *p* value due to large sample size per TriNetX limitation.

## Data Availability

All raw output from data from TriNetX are available by contacting Yilin Song or Gengqing Song.

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
