# Peer review of "Progressive Increase in Small Intestinal Bacterial Overgrowth Risk Following COVID-19 Infection: A Global Population-Based Study"

_diseases, 2025, doi:10.3390/diseases13090275_

Round 1

Reviewer 1 Report

Comments and Suggestions for Authors

This is an interstining study, well designed and  with a high scientific soundness. Although a retrospective study, it is based on a huge audience of Covid-19 suffering patients having developed SIBO post-Covid infection, well-matched with healthy controls; and having a follow-up for 12mo.

My questions/comments are as follows:

  1. I would like to read a paragraph on the way the diagnosis had been established, besides breath test
  2. had these SIBO patients taken any medication to improve the disturbed gut microbiota? a lot of otherwise healthy individuals receive probiotics for immunity enhancement; I suppose much more during Covid-19 pandemic. Is there any information for such an "intervention" which - to our knowledge - would seriously affect [positively] the gut micorbiota. A comment on it, please
  3. Furtheremore, I consider this paper to be an oportunity for an alarm for the uncontrolled use of proton-pump inhibitors [and I am surprised for the difference in "damage" between different PPIs

Author Response

Reviewer 1

1. I would like to read a paragraph on the way the diagnosis had been established, besides breath test

As a shortage of TriNetX, we do not have the exact diagnosis criteria of SIBO each medical center applied other than screen the cases with the ICD-10 code of SIBO (K63.821). While the database enables us to access global-wise patient information, we do not have the luxury of getting detailed information of each case.

2. Have these SIBO patients taken any medication to improve the disturbed gut microbiota? A lot of otherwise healthy individuals receive probiotics for immunity enhancement; I suppose much more during Covid-19 pandemic. Is there any information for such an "intervention" which - to our knowledge - would seriously affect [positively] the gut micorbiota. A comment on it, please

Thank you for this valuable insight. I tried searching in TriNetX database about probiotic as potential medication improving gut microbiota. Unfortunately, there is no ICD-10 code for probiotics established yet. If there is any other medications we can find in the database, it would add to our analysis as a possible confounding factor. I mentioned this content in the limitation part.

3. Furthermore, I consider this paper to be an opportunity for an alarm for the uncontrolled use of proton-pump inhibitors [and I am surprised for the difference in "damage" between different PPIs

Absolutely. We shared the exact same opinion and mentioned in the discussion section. Thank you for pointing this out.

Reviewer 2 Report

Comments and Suggestions for Authors

Dear Author,
thank you for sharing your research.

After a careful review, several points still need clarification:

  1. The use of the TriNetX administrative database allows access to a vast amount of data; however, since this data is originally collected for administrative purposes, it represents an important potential source of bias that should be explicitly acknowledged.

  2. The sample analyzed is remarkably large, and this should be considered in terms of statistical significance. Statistical models may be affected by overly large sample sizes, highlighting significance even where there may be none.

  3. COVID-19 with primarily gastrointestinal manifestations has been described in recent literature (Greco, S., Fabbri, N., Bella, A. et al. COVID-19 inpatients with gastrointestinal onset: sex and care needs’ differences in the district of Ferrara, Italy. BMC Infect Dis 21, 739 (2021). https://doi.org/10.1186/s12879-021-06476-y). Therefore, I would kindly ask you to address this aspect in your manuscript, as it could provide a possible explanation for your long-term findings.

Best regards.

Author Response

Reviewer 2

1. The use of the TriNetX administrative database allows access to a vast amount of data; however, since this data is originally collected for administrative purposes, it represents an important potential source of bias that should be explicitly acknowledged.

Thank you for pointing out this limitation. There are quite a few databases designed for administrative purposes, such as Healthcare Cost and Utilization Project (HCUP), National Practitioner Data Bank (NPDB), Medicare Provider Analysis and Review (MEDPAR).

From the introduction of TriNetX website, it was initially established for assistance of clinical trial design and better research outcomes (Link below). However, with the uncertainty of how data was collected or put in the database, it shares common disadvantages as other databases. I included this part in the limitation paragraph.

https://trinetx.com/about-trinetx/

2.The sample analyzed is remarkably large, and this should be considered in terms of statistical significance. Statistical models may be affected by overly large sample sizes, highlighting significance even where there may be none.

Thank you. While the database enables us to access global patient information, this is one of the key disadvantages of using a large sample size database. I included it in the limitation part.

3. COVID-19 with primarily gastrointestinal manifestations has been described in recent literature (Greco, S., Fabbri, N., Bella, A. et al. COVID-19 inpatients with gastrointestinal onset: sex and care needs’ differences in the district of Ferrara, Italy. BMC Infect Dis 21, 739 (2021). https://doi.org/10.1186/s12879-021-06476-y). Therefore, I would kindly ask you to address this aspect in your manuscript, as it could provide a possible explanation for your long-term findings.

Thank you for directing us to this interesting publication. We included this citation and discussed the possible causes of differences in findings.

Reviewer 3 Report

Comments and Suggestions for Authors

This study aimed to evaluate whether the risk of developing small intestinal bacterial overgrowth (SIBO) increases following COVID-19 infection. Secondary objectives included examining the association between SIBO prevalence and clinical factors such as comorbidities, medication use, and the rate of SIBO diagnosis confirmed by breath testing (BT). A retrospective analysis was conducted using the TriNetX database, involving adult patients (≥18 years) diagnosed with SIBO after COVID-19 infection between January 1, 2022, and May 30, 2024. To control demographic variables and known SIBO risk factors, 1:1 propensity score matching was applied. SIBO incidence over a 12-month period was analyzed using Kaplan-Meier survival curves. The findings revealed a significant increase in SIBO risk post-COVID-19, particularly within the first year after infection, with elevated risk observed across different age groups and among individuals taking certain chronic medications. These results highlight the need for clinical awareness and proactive strategies to address potential long-term gastrointestinal complications.

The present study underscores several important findings: it demonstrates significant gastrointestinal symptoms and gut microbiome disturbances following COVID-19 infection, along with a notable rise in SIBO risk over time. This increase was particularly evident among older adults (especially those over 80 years of age) and individuals with chronic use of proton pump inhibitors (PPIs), opioids, and diphenoxylate. The study also highlights limitations in current diagnostic approaches, such as the underutilization and limited sensitivity of breath tests, emphasizing the need for improved diagnostic strategies, heightened clinical awareness, and more cautious prescribing practices to help mitigate the risk of post-COVID SIBO.

The study presents several strengths. Its use of large-scale, real-world, global data enhances both the relevance and generalizability of the findings. It also stands out as the first quantitative study to investigate the incidence of SIBO following COVID-19 infection. The authors clearly show a progressive increase in SIBO risk over a 12-month period post-infection and effectively identify high-risk populations, including older adults and individuals on chronic medications such as PPIs, opioids, and diphenoxylate. The methodology is rigorous, employing propensity score matching and strict exclusion criteria to reduce confounding factors. Importantly, the study draws attention to the serious complications of untreated SIBO and calls for better clinical management.

However, the study has some limitations. It does not explore in detail the pathophysiological mechanisms underlying the association between COVID-19 and SIBO. While it provides a comprehensive analysis of age-related trends, it does not address potential differences based on sex, ethnicity, or the presence of specific comorbidities. These gaps should be acknowledged and considered for future research.

  • I recommend that these limitations must be included in the study’s limitations section.

Author Response

Reviewer 3

This study aimed to evaluate whether the risk of developing small intestinal bacterial overgrowth (SIBO) increases following COVID-19 infection. Secondary objectives included examining the association between SIBO prevalence and clinical factors such as comorbidities, medication use, and the rate of SIBO diagnosis confirmed by breath testing (BT). A retrospective analysis was conducted using the TriNetX database, involving adult patients (≥18 years) diagnosed with SIBO after COVID-19 infection between January 1, 2022, and May 30, 2024. To control demographic variables and known SIBO risk factors, 1:1 propensity score matching was applied. SIBO incidence over a 12-month period was analyzed using Kaplan-Meier survival curves. The findings revealed a significant increase in SIBO risk post-COVID-19, particularly within the first year after infection, with elevated risk observed across different age groups and among individuals taking certain chronic medications. These results highlight the need for clinical awareness and proactive strategies to address potential long-term gastrointestinal complications.

The present study underscores several important findings: it demonstrates significant gastrointestinal symptoms and gut microbiome disturbances following COVID-19 infection, along with a notable rise in SIBO risk over time. This increase was particularly evident among older adults (especially those over 80 years of age) and individuals with chronic use of proton pump inhibitors (PPIs), opioids, and diphenoxylate. The study also highlights limitations in current diagnostic approaches, such as the underutilization and limited sensitivity of breath tests, emphasizing the need for improved diagnostic strategies, heightened clinical awareness, and more cautious prescribing practices to help mitigate the risk of post-COVID SIBO.

The study presents several strengths. Its use of large-scale, real-world, global data enhances both the relevance and generalizability of the findings. It also stands out as the first quantitative study to investigate the incidence of SIBO following COVID-19 infection. The authors clearly show a progressive increase in SIBO risk over a 12-month period post-infection and effectively identify high-risk populations, including older adults and individuals on chronic medications such as PPIs, opioids, and diphenoxylate. The methodology is rigorous, employing propensity score matching and strict exclusion criteria to reduce confounding factors. Importantly, the study draws attention to the serious complications of untreated SIBO and calls for better clinical management.

However, the study has some limitations. It does not explore in detail the pathophysiological mechanisms underlying the association between COVID-19 and SIBO. While it provides a comprehensive analysis of age-related trends, it does not address potential differences based on sex, ethnicity, or the presence of specific comorbidities. These gaps should be acknowledged and considered for future research.

Thank you for the thorough interpretation and positive feedback on the strengths of our manuscript. There are two main mechanisms which explain the possible causes COVID-19 infection leading to SIBO prevalence. The first one is the dysfunction of RAS system from COVID-19 virus binding to ACE-2 receptor in the small intestine epithelial cells. The second one is based on the interruption of gut microbiota balance by COVID-19 virus, leading to gastrointestinal motility disorders. (Line 56-61)

We took into consideration the potential differences of sex, ethnicity and related comorbidities as confounding factors of the SIBO prevalence outcome. We included these features/factors as the factor for propensity score match (Line 109-113).

Hope the explanations address your concerns.

Round 2

Reviewer 1 Report

Comments and Suggestions for Authors

thank you for your response

No more comments